# Genetic and Phenotypic Evaluation of European Maize Landraces as a Tool for Conservation and Valorization of Agrobiodiversity

**DOI:** 10.3390/biology13060454

**Published:** 2024-06-19

**Authors:** Carlotta Balconi, Agustin Galaretto, Rosa Ana Malvar, Stéphane D. Nicolas, Rita Redaelli, Violeta Andjelkovic, Pedro Revilla, Cyril Bauland, Brigitte Gouesnard, Ana Butron, Alessio Torri, Ana Maria Barata, Natalija Kravic, Valérie Combes, Pedro Mendes-Moreira, Danela Murariu, Hrvoje Šarčević, Beate Schierscher-Viret, Morgane Vincent, Anne Zanetto, Bettina Kessel, Delphine Madur, Tristan Mary-Huard, André Pereira, Domnica Daniela Placinta, Alexandre Strigens, Alain Charcosset, Sandra Goritschnig

**Affiliations:** 1CREA—Council for Agricultural Research and Economics, Research Centre for Cereal and Industrial Crops, via Stezzano 24, 24126 Bergamo, Italy; rita.redaelli@crea.gov.it (R.R.); alessio.torri@crea.gov.it (A.T.); 2INRAE, CNRS, AgroParisTech, GQE—Le Moulon, Université Paris-Saclay, 12 route 128, 91190 Gif-sur-Yvette, France; agustin-oscar.galaretto@inrae.fr (A.G.); stephane.nicolas@inrae.fr (S.D.N.); cyril.bauland@inrae.fr (C.B.); valerie.combes@inrae.fr (V.C.); delphine.madur@inrae.fr (D.M.); tristan.mary-huard@agroparistech.fr (T.M.-H.); alain.charcosset@inrae.fr (A.C.); 3Misión Biológica de Galicia Consejo Superior de Investigaciones Científicas, Pazo de Salcedo Carballeira, 8 Salcedo, 36143 Pontevedra, Spain; rmalvar@mbg.csic.es (R.A.M.);; 4Maize Research Institute Zemun Polje, 11000 Belgrade, Serbia; violeta@mrizp.rs (V.A.); nkravic@mrizp.rs (N.K.); 5UMR AGAP Institut, CIRAD, INRAE, Institut Agro, University Montpellier, F-34398 Montpellier, Francemorgane.vincent@inrae.fr (M.V.); anne.zanetto12@gmail.com (A.Z.); 6Banco Português de Germoplasma Vegetal, Quinta de S. José, S.Pedro de Merelim, 4700-859 Braga, Portugal; anamaria.barata@iniav.pt; 7Coimbra School of Agriculture, Polytechnic University of Coimbra (ESAC-IPC), 3045-093 Coimbra, Portugal; pmm@esac.pt (P.M.-M.); andre.pereira@esac.pt (A.P.); 8CERNAS—Research Centre for Natural Resources, Environment and Society, Bencanta, 3045-601 Coimbra, Portugal; 9Suceava Genebank, B-Dul. 1 Mai 17, 720224 Suceava, Romania; danela.murariu@svgenebank.ro (D.M.); domnica.placinta@svgenebank.ro (D.D.P.); 10Faculty of Agriculture, University of Zagreb, Svetošimunska Cesta 25, 10000 Zagreb, Croatia; hsarcevic@agr.hr; 11Agroscope, Route de Duillier 60, 1260 Nyon, Switzerland; beate.schierscher-viret@agroscope.admin.ch; 12KWS SAAT SE & Co. KGaA, Grimsehlstr. 31, 37574 Einbeck, Germany; bettina.kessel@kws.com; 13INRAE, UMR MIA Paris-Saclay, Université Paris-Saclay, AgroParisTech, 91120 Paris, France; 14DSP—Delley Semences et Plantes SA, Route de Portalban 40, 1567 Delley, Switzerland; strigens@dsp-delley.ch; 15ECPGR, Alliance of Bioversity International and CIAT, Via di San Domenico 1, 00153 Rome, Italy

**Keywords:** maize, genebanks, evaluation network, conservation, genetic groups, genetic diversity, phenological and morphological traits, genetic resources

## Abstract

**Simple Summary:**

Maize is one of the major crops of the world for feed, food, and industrial uses. It originated in Central America and was first introduced into Europe at the end of the 15th century. Due to its adaptability, farmers and breeders across Europe have developed a wide diversity of local maize varieties with different characteristics over the past centuries. Many of these are conserved in genebanks’ seed collections, but little is known about their specific characteristics. Here, we present results obtained by the European Evaluation Network for Maize, a private–public partnership with partners from nine countries aimed at promoting the valorization of maize genetic resources in breeding programs. The work describes the selection and the genetic and phenotypic evaluation of a collection of 626 maize landraces preserved in European genebanks, providing evidence for historic introductions and geographic adaptation. In a world where climate change, rising food prices, and other issues are affecting food security and the environment, the conservation and use of crop diversity is becoming increasingly important. The results of our study will facilitate the use of maize genetic resources in breeding for resilience to climate change, for sustainable agriculture, food security, and food quality.

**Abstract:**

The ECPGR European Evaluation Network (EVA) for Maize involves genebanks, research institutions, and private breeding companies from nine countries focusing on the valorization of maize genetic resources across Europe. This study describes a diverse collection of 626 local landraces and traditional varieties of maize (*Zea mays* L.) from nine European genebanks, including criteria for selection of the collection and its genetic and phenotypic diversity. High-throughput pool genotyping grouped the landraces into nine genetic groups with a threshold of 0.6 admixture, while 277 accessions were designated admixed and likely to have resulted from previous breeding activities. The grouping correlated well with the geographic origins of the collection, also reflecting the various pathways of introduction of maize to Europe. Phenotypic evaluations of 588 accessions for flowering time and plant architecture in multilocation trials over three years confirmed the great diversity within the collection, although phenotypic clusters only partially correlated with the genetic grouping. The EVA approach promotes conservation of genetic resources and opens an opportunity to increase genetic variability for developing improved varieties and populations for farmers, with better adaptation to specific environments and greater tolerance to various stresses. As such, the EVA maize collection provides valuable sources of diversity for facing climate change due to the varieties’ local adaptation.

## 1. Introduction

Maize (*Zea mays* L.) was brought to Europe for the first time in 1494, when Columbus arrived at Seville in Southern Spain [1]. Successive introductions of maize were made in Southwestern Europe. Genotyping analyses [2] have confirmed that maize was first introduced in Southern Spain from the Caribbean area, and shortly after from North America. The most likely hypotheses for this second introduction are that it arrived through the European Atlantic coast [1] and French explorations conducted by Giovanni da Verrazzano. From these two introductions originated two main European maize pools, the Mediterranean and European flints, which were crossed afterwards and selected for adaptation to diverse environments and uses. 

Plant genetic resources for food and agriculture (PGRFA) are the backbone of the world’s food security as they provide the necessary genetic diversity for researchers, breeders, and farmers to continuously develop new varieties that can withstand challenges associated with the climate crisis and support sustainable agricultural production. In view of the changing climate, enhanced efforts are needed to conserve existing locally adapted PGRFA diversity and make it accessible for breeding programs through characterization. 

The European Cooperative Programme for Plant Genetic Resources (ECPGR) is a pan-European network active since 1980, connecting national programs on PGRFA through expert working groups and other initiatives with the aim to support conservation and sustainable utilization of PGRFA, in line with the Plant Genetic Resources Strategy for Europe [3]. An ECPGR initiative launched in 2019, the European Evaluation Network (EVA) brings together national genebanks and other stakeholders from the public and private sectors into public–private partnerships (PPPs) jointly working on creating and sharing knowledge on PGRFA conserved in European genebanks [4]. Using standardized protocols, crop-specific EVA networks evaluate genebank materials, which are often understudied, in multiple locations across Europe, thereby identifying climate-resilient breeding material and enriching genebank inventories, ultimately promoting their sustainable use. The EVA Maize Network, launched in 2020, currently has partners from nine countries, including eleven genebanks and research institutes and eight breeding companies (https://www.ecpgr.cgiar.org/eva/eva-networks/maize, accessed on 14 April 2024). It builds on existing national networks, connecting members of the ECPGR Maize Working Group with breeding companies involved in the French PPP initiative ProMaïs (http://pro-mais.org/, accessed on 14 April 2024). Network activities are focused on the valorization of maize genetic resources across Europe, with a focus on landraces and traditional varieties [5], which are valuable sources of genetic diversity for facing climate change, due to their local adaptation.

The EVA Maize Network also links to a previous initiative, the European program RESGEN88 (1997–2001, https://www.agap-ge2pop.org/eu-genres-088/, accessed on 14 April 2024). This project aimed to obtain a better knowledge of the maize landraces grown in Europe in the past centuries and now maintained in genebanks, and to define the best conditions for their valorization and exploitation. The RESGEN88 program involved seven countries: the Netherlands, Germany, Greece, France, Portugal, Italy, and Spain and shared a set of 394 local maize populations. Each country defined a core set of landraces representative of their national maize collection. A preliminary evaluation through agronomical trials, collection of morphological descriptors, and molecular analyses via RFLP [6] resulted in the constitution of a European Maize Landraces Core Collection (EUMLCC) of 96 populations, which synthesized the genetic variability present in the original set of genotypes [1,7]. Evaluations of the EUMLCC by partners explored grain composition through near-infrared (NIR) spectroscopy, revealing large variability especially of carotenoid compounds [8], as well as resistance and tolerance to corn borer attack [9] and tolerance to cold conditions. 

In this paper, we present the concept applying PPP to the multilocation evaluation of European maize genetic resources, including part of the EUMLCC. We describe the contributing maize collections and criteria for selection of interesting landraces and present results of diversity studies based on genotypic data and phenotypic traits. We show how the EVA approach provides a model for effective collective generation of knowledge on PGRFA, making them more accessible for use in research and breeding.

## 2. Materials and Methods

### 2.1. Plant Material 

A total of 626 different European maize landrace populations provided by nine genebank partners in the project were selected for inclusion in the EVA collection. Criteria considered for selection are described in detail in the results. Accessions were regenerated by holding institutes following an established protocol [10] and distributed with SMTA under the terms of the Multilateral System of the International Treaty for Plant Genetic Resources for Food and Agriculture (ITPGRFA) to partners for evaluations. 

A set of five common registered check hybrids from company partners representing FAO maturity groups (210-300-400-500-600) was included in evaluations to enable cross-comparison of the different trials. Table 1 summarizes the EVA Maize collection and Appendix A lists the passport data of all accessions.

### 2.2. Genotyping

For genotyping of the 626 landraces in the EVA maize collection, each landrace population was represented through randomly sampling 15 individual plants from which equal amounts of leaves were mixed together prior to DNA extraction, to obtain one DNA pool per landrace as described previously [26]. Plants were germinated directly from genebank seeds, with no prior multiplication. 

The DNA pools of the 626 landraces were genotyped using the 50K Illumina Infinium HD array [27] according to manufacturer’s instructions (Illumina Inc., San Diego, CA, USA). SNPs were filtered based on their suitability for diversity analysis and their quality for predicting allelic frequency in DNA pools, following a procedure described previously [26], resulting in a final set of 23,412 SNPs for further analysis. 

Fluorescence data for alleles A and B of the 23,412 SNPs were extracted for each landrace with GenomeStudio’s Genotyping Module software (v2010.2, Illumina Inc., San Diego, CA, USA) and used for predicting allelic frequency using a freely available script (https://doi.org/10.15454/GANJ7J, accessed on 14 April 2024), following the two-step procedure described in [26] based on the fluorescence intensity ratio (FIR) of alleles A and B for each SNP. First, we tested whether SNPs were monomorphic or polymorphic for each landrace. For SNPs that were polymorphic within each landrace, we then estimated the allelic frequency of the B allele using a generalized linear model calibrated on FIR data from 1000 SNPs from two series of controlled pools (see [26] for more detail and Equation (2) for the model). This two-step approach led to a global mean absolute error of 3% and was more conservative for SNPs that were fixed or close to fixation than for SNPs with balanced allelic frequency [26]. The threshold to reject the hypothesis that landraces were monomorphic was set to 5%, indicating that 5% of landraces were expected to be declared polymorphic while actually being monomorphic (false positive). 

### 2.3. Diversity Analysis

#### 2.3.1. Estimation of Genetic Diversity Parameters

For each landrace, we averaged across 23,412 SNPs the mean allele number (A), the minor allele frequency (MAF), and the expected heterozygosity (H) [28,29]. Considering that all landraces were represented by 15 different plants (30 gametes), we did not apply a correction for the number of individuals when estimating these parameters because it would have resulted in only a small increase for diversity parameters (3.4% according to [30]). We also estimated genetic differentiation (G_st_) between individual landraces using the 23,412 SNPs, according to [28]. We then extended the diversity measures at the complete panel level and for specific genetic groups (described below), using locus average allelic frequency. For genetic groups, only highly assigned landraces (>0.6) were considered.

#### 2.3.2. Genetic Structure and Relationship between Landraces 

We estimated the genetic distance between all landraces using modified Roger’s distance (MRD) [31] with ad hoc scripts, and fixation index (F_ST_) based on the allelic frequencies of 23,412 SNPs using R package HFst [32]. To decipher the structure of genetic diversity within our panel of 626 landraces and their relations with worldwide diversity previously described in [33], we used three approaches:(1)A distance-based approach using MRDs between the 626 landraces to perform principal coordinate analysis (PCoA) [34], as well as hierarchical clustering. We applied either Ward or neighbor-joining algorithms using the “hc” and “bionj” functions of “ape” R package v 5.0 [35], respectively; (2)A Bayesian multi-locus approach, implemented in the ADMIXTURE v1.3.0 software, to assign probabilistically each landrace to K ancestral populations assumed to be in Hardy–Weinberg equilibrium [36]. Different methods were used to identify the most appropriate number of ancestral populations (K): cross-validation error or difference between successive cross-validations [36] and graphical methods [37]. Since ADMIXTURE requires multi-locus genotypes of individual plants, we simulated the genotype of five individuals for each population for a subset of 2500 independent SNPs to avoid artifacts of linkage disequilibrium (See Method S5 in [33] for more details).(3)A linear penalized regression approach, to quantitatively assign each of the 626 landraces to seven genetic groups established for 156 landraces representing European and American diversity [33]. Allelic frequencies at 23,412 SNPs of each of the seven genetic groups were estimated considering landraces with a membership superior to 0.6. Admixture coefficients associated with the new landraces were then obtained through a penalized regression approach via fitting, for each of the 626 landraces, a linear regression model using landrace allelic frequencies as response variables and group frequencies as explanatory variables, using the R package “quadprog” [38] v1.5-8. The coefficients of each regression were constrained to be positive and sum to one and were considered as estimates of the admixture coefficients.

We used R package “ggtree” [39] to draw the dendrograms derived from hierarchical clustering. Barplots from admixture analysis and penalized regression as well as the color scales for Hs, DS, and PH were added using “ggtreeExtra” R package [40,41].

Finally, to investigate the relationship between admixture and geographical origins, we mapped individual pie plots representing the admixture coefficients of accessions to their collection sites, using “ggplot2” [42] and “scatterpie” [43] R packages.

### 2.4. Phenotypic Evaluations

Among the 626 landraces that were genotyped, 38 could not be phenotyped due to limited seed availability. The 588 other landraces were evaluated by project partners in eleven locations in three sets of 175–218 accessions per set, over three years. Each set was evaluated on at least two locations in one growing season, yielding a total of 22 environments (Table 2). Each accession set covered maturity ratings from FAO100 to 800 and was distributed across locations to match local capacities and growing conditions, resulting in 2–9 trial locations per accession (Appendix A). Evaluations followed a standard experimental protocol, using standard agronomic practices. The protocol consisted of evaluating a set of landraces along with a set of check hybrids, but the number of repetitions and the number of check hybrids varied in the different trials. 

Data were collected for phenology traits, including days to tasseling [DT] and days to silking [DS] and for plant architecture traits, including plant height [PH] and ear height [EH], following standard protocols and using standardized scoring scales (Table 3) and were stored in a project specific database. Derived traits including anthesis–silking interval (ASI = DS-DT) and relative ear height (EPHR) were calculated from the raw data.

### 2.5. Statistical Analysis of Phenotypic Data

A combined analysis of variance was performed for the phenotypic traits studied using the PROC MIXED procedure in SAS software v. 9.4 (SAS Institute Inc.: Cary, NC, USA). The best linear unbiased estimators (BLUEs) for each variety were calculated based on pooled data for all environments. Varieties were considered as fixed effects while replication (trials) was considered as random. 

A principal component analysis was carried out on the BLUEs of the 588 landraces and 5 check hybrids using the six evaluated traits (Table 3). The data were standardized (mean = 0 and standard deviation = 1) to avoid scaling problems. Principal components with an eigenvalue > 1 became new variables, with which cluster analysis was carried out. For cluster analysis, we first used the FASTCLUS procedure of SAS to perform preliminary clustering using Euclidean distance (the recommended method for more than 100 varieties), choosing 15 preliminary clusters based on the pseudo F statistic. Subsequently, we performed the CLUSTER procedure using average linkage as a clustering method. Finally, the results of the hierarchical clustering analysis were displayed as a dendrogram using the TREE procedure (SAS Institute Inc.: Cary, NC, USA).

## 3. Results

### 3.1. Creating the EVA Maize Collection

The EVA maize collection included materials provided by genebank partners from Croatia, France, Italy, Portugal, Romania, Serbia, Spain, and Switzerland, reaching a total of 626 accessions, originating from 26 countries, and focusing on locally adapted landraces of various FAO maturity groups (Appendix A). These constituted key accessions for each holding institute (see Appendix A for details about each genebank). Four genebanks previously involved in the RESGEN88 project contributed 75 accessions belonging to the EUMLCC core collection. The selection processes adopted by each genebank to include accessions into the EVA maize collection are described below and summarized in Table 1.

**Croatia—University of Zagreb, Faculty of Agriculture (HRV041):** The subset of 50 landraces from the collection of the University of Zagreb Faculty of Agriculture (Croatian genebank) included in the EVA maize collection was selected to represent the collection based on its wide geographical distribution as well as on previous characterization data [11]. It included accessions from the continental and Mediterranean regions of Croatia with a wide range of variation in phenological and morphological traits such as maturity group, plant height, ear length, and ear width.

**France—INRAE—Montpellier (FRA015):** The CRB GAMéT genebank made available for the EVA maize collection the French representative collection (RMNC, maintained with the Promaïs partnership), initially defined for the RESGEN88 project. A set of 80 landraces (16 of them inserted in EUMLCC) were chosen based on morphological traits using Mstrat software (v 4.1) to maximize the number of phenotypic classes with at least one population. The morphological traits considered for selection were measured and analyzed in the classification of French maize populations [7,12]. 

**Italy—CREA—Cereal and Industrial Crops, Bergamo (ITA386):** The set of 65 landraces shared by CREA for the EVA maize collection included 19 varieties of the Italian collection belonging to the EUMLCC selected in the frame of RESGEN88 [8]. Additionally, accessions originally collected around 1950 from different Italian areas were chosen. The Italian landraces selected for the EVA maize collection have also recently been included in different programs devoted to valorization of local genetic resources and their possible utilization as sources of stress-tolerance traits in pre-breeding activities [13].

**Portugal—INIAV, Braga (PRT001):** The set of 42 Portuguese landraces shared by INIAV for the EVA project included 17 varieties of the Portuguese collection belonging to the EUMLCC. Twenty-five additional accessions were chosen from within the frame of the national collection of 70 varieties based on morphological, agronomic, and molecular characterization, as well as on previous knowledge from national breeding programs [14,15].

**Portugal—ESAC-IPC, Coimbra (PRT053):** The six Portuguese landraces shared by the Escola Superior Agrária de Coimbra (ESAC, Portugal) in the EVA maize collection included accessions collected during a collecting mission focused on maize varieties for traditional bread production, and others were tested in drought trials [16]. Additionally, some accessions were selected based on previous studies, in which yield evaluations were conducted at different altitudes. The ‘Bilhó’ OPV population (white flint maize) was selected at higher-altitude eco-geographic areas, generally with a lack of water, in contrast with ‘Sangalhos’ from low altitudes.

**Romania—Suceava Genebank (ROM007):** The maize collection in the Suceava Genebank represents 32% of the entire Romanian national collection. Most of the 6143 maize accessions in the national collection are local landraces from Romania (60%), especially from mountainous and sub-mountainous areas (up to 400 m altitude). The 51 maize landraces included in the EVA maize collection were selected from among early and semi-early accessions (FAO groups 200–300), collected between 20 and 50 years ago (1973–2000), with resistance to low temperatures [17,18]. 

**Serbia—Maize Research Institute Zemun Polje (SRB001):** The criteria followed for selection of 91 accessions by MRIZP, Serbia, included drought tolerance observed in a two-year screening program under a managed stress environment in Egypt, stability and high grain yield in a two-year trial under rain-fed conditions and with higher plant density applied under temperate conditions at Zemum Polje, and good performance regarding traits important for breeding [19,20], representing valuable donors of genes responsible for higher grain yield in crossings with germplasm from different heterotic groups. The selection included mainly accessions from the Western Balkans but also some exotic materials that may be a valuable source for drought tolerance. 

**Spain—Misión Biológica de Galicia, Pontevedra (MBG-CSIC)—(ESP004, ESP007, ESP009, ESP016, ESP019):** Among the 137 Spanish landraces included in the EVA maize collection, 23 landraces were also contained within the EUMLCC. As Spanish germplasm is very variable, all Spanish landraces with enough seed availability were included. Previous knowledge on corn borer resistance, drought and cold tolerance or suitability for making bread was available for many of them [1,9,21]. Pontevedra (ESP009) coordinated the distribution of the accessions from other Spanish holding institutes. 

**Switzerland—Agroscope, Nyon (CHE001):** The 86 accessions included in the EVA maize collection are Swiss and Liechtenstein maize landraces selected to represent different collecting phases as well as different regions of Switzerland. The accessions have phenological and morphological data such as plant height, ear length, kernel color, and others [22,23,24,25].

The EVA maize collection of 626 European landraces was distributed to EVA Maize Network partners for field evaluations and genotyping, with the aim to assess the genetic diversity available in the collection and to evaluate the landraces in multilocation trials for phenotypic characteristics important for further use in breeding and research.

### 3.2. Genotypic Diversity of the Collection

Out of 23,412 useful SNPs, 4 were monomorphic for the whole EVA panel. Minor allelic frequency (MAF) was 0.25 (± 0.14) (Table 4). At group level the average number of alleles per locus was 2.00 (± 0.01), while within landraces it was 1.71 (± 0.13) and ranged from 1.23 (SRB0772, Zuti Poluzuban) to 1.92 (HRV0770, Domaci Osmak) (see Appendix A). The mean total expected heterozygosity among (H_t_) and within landraces (H_s_) in the whole panel were 0.33 (± 0.15) and 0.21 (± 0.05), respectively (Table 4), leading to a mean genetic differentiation between landraces (G_st_) of 0.35 (± 0.14) across the whole panel. The expected heterozygosity within landraces varied from 0.03 (SRB0772, Zuti poluzuban) to 0.30 (ITA0083, Giallo Tosoratti) (Appendix A, Appendix A). Only 13 landraces had an H_s_ below 0.10, suggesting that these landraces were either not well maintained in the genebank or that they had been under strong bottleneck during their initial collection.

We analyzed the genetic structure of 626 landraces using the ADMIXTURE program [36]. Likelihood analysis of ADMIXTURE results indicated that the optimal numbers of genetic groups were K = 2, K = 4, K = 6, and K = 9 (Appendix A). We compared this new genetic structuration for successive K values with the assignment to 7 genetic groups determined in [33], using a penalized regression approach (Appendix A, Appendix A). At K = 2, “Northern Flint” landraces split from other landraces, which is highly consistent with the first axis of the PCoA analysis (Figure 1) and the first split in the dendrogram (Figure 2). At K = 4, genetic groups corresponded to those identified in Europe [33]: Northern Flint, Pyrenean–Galician Flint, Italian Flint, and Corn Belt Dent. At K = 6, the groups of Northern Flint and Italian Flint were conserved, while the Pyrenean–Galician group was split into two equal genetic groups corresponding to collections from humid Northern Spain (landraces from Galicia) and French/Spanish landraces from the Pyrenees. Finally, a small group of popcorn landraces from Spain, Romania, and France was separated from the group composed of Corn Belt Dent landraces. At K = 9, the groups Northern Flint (K8), Corn Belt Dent (K4), popcorn (K1), and Pyrenean (K9) were maintained. On the other hand, Portuguese landraces, which were considered an admixture between Pyrenean–Galician, Andean, and Northern Flint Groups [45], became a new genetic group (K6), closely related to the genetic group of Galician landraces (K7). Similarly, the Italian Flint group [45] was split into two equal genetic groups corresponding to landraces from Northern Italy (K5) and South–Central Italy (K3). Interestingly, landraces collected along the Adriatic coast of Croatia and Montenegro were strongly assigned to the South–Central Italy (K3) genetic group. In addition, Southern dry Spanish landraces, which penalized regression showed to be an admixture between Caribbean and Italian Flint groups [45], became a specific genetic group (K2).

The new genetic structuration with K = 9 was highly consistent with our previous analysis based on 156 worldwide landraces [33] and with the history of both introductions of maize in Europe, the first from the Caribbean into Southern Spain and the second from North America into Northern Europe. In addition, it allowed us to go deeper in the understanding of spatial organization of the genetic diversity along the European continent. For further analysis, we focused on this last structure of ancestry.

The size of these nine genetic groups varied from 14 (K1, K6) to 75 (K8) landraces (Table 4). We compared the diversity parameters of landraces assigned to nine genetic groups, considering only landraces with a membership above 0.6 (Appendix A, Table 4). The H_t_ in the nine groups varied from 0.22 (K8, Northern Flint) to 0.33 (K4, Corn Belt Dent), with the latter value equivalent to the whole panel. The mean within-gene diversity in each group (H_s_) varied from 0.14 (K8, Northern Flint) to 0.24 (K4, Corn Belt Dent). Average genetic distance (MRD) between landraces in each group ranged from 0.16 (K6, Portuguese and K7, Galician) to 0.23 (K2, Southern Spain). The number of monomorphic markers in each group varied from 956 in group K1 (popcorn) to 81 in group K4 (Corn Belt Dent). Genetic differentiation (G_st_) within genetic groups varied from 0.17 (K6, Portuguese) to 0.36 (K2, Southern Spain and K8, Northern Flint). The estimation of fixation index (F_ST_) between genetic groups added information on their relationship (Appendix A). The largest F_ST_ (0.32) was found between the group of K2 (Southern Spain) and K8 (Northern Flint) while the least differentiated genetic groups were K7 (Galician) and K9 (Pyrenean)—0.05. In addition, the genetic group K6 (Portuguese) was also weakly differentiated from these two groups. Little differentiation was found between landraces from groups K5 (Northern Italian) and K3 (South–Central Italian; 0.08) or between them and group K2 (Southern Spain; 0.10 and 0.09, respectively). Finally, K4 (Corn Belt Dent) and K6 (Portuguese) groups were weakly differentiated (0.10).

To better understand the relationships between landraces and the nine genetic groups, we performed PCoA analysis based on MRD distance (Figure 1) and clustering analysis using both the Ward algorithm based on MRD distance (Figure 2) and neighbor joining based on F_ST_ (Appendix A).

In Figure 1, the first axis (PC1, 13.9% of total variation) discriminated landraces from Northern Europe (Switzerland and some landraces from Northeastern France) corresponding to Northern Flint landraces (K8) against landraces from Southern Spain (K2). This was consistent with the two groups identified with the ADMIXTURE analysis at K = 2 and corresponded to the first split in the dendrogram tree (Figure 2). The second axis (PC2, 7.27% of total variation) separated French, Spanish, and Portuguese landraces collected in the regions of the Pyrenees and Galicia (K6, K7, K9) from Corn Belt Dent and popcorn-like landraces from France, Romania, and Southern Spain (K1 and K4 genetic groups). Interestingly, flint landraces from Northern and South–Central Italy were intermediate between Southern Spanish landraces and Pyrenean–Galician–Portuguese Flint landrace groups (K6, K7, K9) on the first two axes of the PCoA. The third axis (PC3, 4.72% of total variation) separated Spanish and Galician Flint landraces from Northern and South–Central Italian Flint landraces (K3 and K5). The fourth PCA axis (PC4, 1.9% of total variation) separated popcorn landraces (K1) from the rest of the panel (Appendix A).

Dendrogram trees based on Ward clustering (Figure 2) or neighbor joining (Appendix A) provided a finer view of the relationship between the clustering and ancestry of the landraces. At a higher level of the dendrogram tree, landraces from Switzerland and Northern France (K8) assigned to the Northern Flint Arca group [33] were split from all other landraces/genetic groups, following the PCoA first axis and ADMIXTURE groups at K = 2. Accordingly, the Northern Flint group displayed the highest differentiation from landraces from five other genetic groups (K1, K2, K3, K4, K5 with F_ST_ ranging from 0.19 to 0.32 in Appendix A). Northern Flint landraces were closer to K6 (Portuguese, F_ST_ = 0.14), K7 (Galician, F_ST_ = 0.21), and K9 (Pyrenean, F_ST_ = 0.17). This was expected since the Pyrenean–Galician varieties resulted from admixture events between Northern Flint from Northern Europe and Southern Spanish landraces from Caribbean origins [46]. The landraces were then divided into two distinct groups in the tree: (i) landraces from the Pyrenees, Galicia, and Portugal (K6, K7, K9) corresponding to the Pyrenean–Galician Arca group [33], which were the least differentiated groups (F_ST_ < 0.07 in Appendix A), and (ii) landraces from Italy, Spain, and Southeastern Europe (K2, K3, K4) corresponding to Italian Flint and Corn Belt Dent Arca groups, respectively. At a lower level, the Pyrenean–Galician–Portuguese landraces split in different subgroups: (i) a first cluster of pure French–Pyrenean landraces (K9) with admixed French landraces with Northern Flint or Corn Belt Dent and admixed Spanish–Galician landraces; and (ii) a second cluster of Portuguese (K6) and Galician landraces (K7) with some admixed landraces between these two groups, originating from Northern Portugal. The Pyrenean–Galician–Portugal cluster was therefore mainly composed of three genetic groups: K6 (mainly Portuguese landraces), K7 (Spanish–Galician landraces), and K9 (mainly landraces collected in the French Pyrenees). Interestingly, some subgroups were admixed almost exclusively between these or with K4 (Corn Belt Dent) and K8 (Northern Flint).

On the other branches of the tree, the landraces were divided into two major groups: (i) landraces highly or partially assigned to the Italian Flint Arca group (or admixed with it), and (ii) landraces mainly from Southeastern Europe assigned in great part to the Corn Belt Dent Arca group. The Italian Flint Arca group was split in three subclusters: (i) one subcluster of landraces from Northern Italy and Southern Switzerland, that were purely assigned to Italian Flint (K5); (ii) a new subcluster of landraces from South–Central Italy and from Croatia along the Adriatic coast (K3), and (iii) a third subcluster of landraces from Southern Spain, cultivated in dry areas (K2), that appeared as admixed between Italian Flint and the Caribbean Arca group. A small subcluster presented a level of admixture with the K1 genetic group and was composed of popcorn varieties. The second subcluster, partially assigned to Italian Flint, included highly admixed landraces, also not clearly assigned to any ancestral population. It was mainly composed of landraces collected in Spain, Italy, France, and Croatia. 

Finally, the cluster highly assigned to Corn Belt Dent was made up of three clearly separated subclusters. The first one was a small group including popcorn varieties mainly from Spain and identified as a separate genetic group through the ADMIXTURE analysis (K1). The second one was composed of accessions highly assigned to the Corn Belt Dent ancestral group and K4 genetic group. This group included landraces mainly collected in Croatia, Serbia, and Bosnia. The third subcluster was composed of admixed landraces mainly from Romania, Croatia, and Bulgaria. Landraces in this group were partially assigned to a mixture of most ancestral groups, except for Caribbean. In terms of ADMIXTURE results, most landraces showed different levels of admixture between K4, K8, and K9.

The organization of the neighbor-joining dendrogram based on F_ST_ was almost identical, except that the Corn Belt Dent (K4) cluster was closer to the Northern Flint genetic group (K8) (Appendix A). In the same direction, a set of subclusters of mainly Romanian landraces were placed between Northern Flint and Corn Belt Dent landraces following a gradual decrease in Northern Flint ancestry. Specifically, the subcluster integrated with admixed Pyrenean–Northern Flint landraces, placed close to Pyrenean in the Ward dendrogram, was moved next to the Northern Flint landraces.

To analyze the spatial structure of these genetic groups, we mapped the admixture coefficients of the nine genetic groups for 483 landraces for which GPS coordinates of collection sites were available (Figure 3). Landraces belonging to the nine genetic groups clustered according to their geographic origins except for popcorn landraces that spread from Southern Spain to Southwestern France. The genetic structure matched very well with eco-geographical regions corresponding to various environments encountered in Europe, suggesting that landraces of these different genetic groups corresponded to adaptation to specific environments. For instance, Corn Belt Dent landraces (K4) were mapped mainly in continental climate areas in Southeastern Europe with hot summers, while Northern Flint landraces (K8) were cultivated in the North of Europe and well adapted to cold temperatures during spring sowings. Landraces from Spain were split into two groups corresponding to two contrasted environments: humid oceanic climate in Galicia vs. hot and dry Mediterranean climate in Southern Spain. Contrasting environments could also explain the shift into different genetic groups between Pyrenean and Galician (transition from the Mediterranean to Atlantic/Lusitanian region), as well as between Galician and Portuguese landraces (transition from the Atlantic to Mediterranean region). Landraces from Southern Spain appeared as the closest relatives to landraces from the Canary Islands that were assigned to the Caribbean Arca group (Appendix A). In addition, cultural or historical processes may have contributed to the geographic distribution of genetic groups. This is likely to have been the case for a group of Swiss landraces highly assigned to the K5 group that were collected in the Ticino region, known for being in close relationship with Italian culture. The geographic structure was also evident when looking at different K values detected in the ADMIXTURE analysis (Appendix A). Interestingly, at K = 4 and K = 6, landraces from the regions of Southern Spain and Southern Italy mainly belonged to the same genetic groups (K1 and K3, respectively).

### 3.3. Phenotypic Diversity of the Collection

Based on seed availability and field capacity for evaluations, 588 out of the 626 landraces were evaluated in field trials. Five commercial hybrids of different FAO cycles (from 210 to 600) were included in the trials as checks to find out which landraces were best suited to specific environments. To assess the phenotypic diversity of the collection, we analyzed phenological traits including days to silking (DS), days to tasseling/anthesis (DT), and anthesis–silking interval (ASI) as well as plant architecture traits including plant height (PH), ear height (EH), and relative ear height (EPHR); mean data were based on data recorded in 2–9 locations per accession (Appendix A). These traits were chosen for preliminary phenotypic description of the European landraces because they provide key information for adaptation to different environments. BLUEs for each of the traits studied and for all landraces and checks can be found in Appendix A.

The great amplitude of range of variation showed that there was large variability among landraces for all traits studied (Table 5). Regarding male and female flowering, about 25% of the landraces were earlier than the earliest FAO210 hybrid check. Also, there were landraces that flowered later than the latest hybrid FAO600. In terms of plant and ear height, 75% and 50% of the landraces, respectively, were shorter than any check hybrid but some landraces were also taller than the checks. In terms of ASI, the wide range of variation of the landraces contrasted with that of the hybrids as did the ratio between plant and ear heights.

The wide range of variation in the EVA collection for female flowering and plant height is shown in Figure 4 and Appendix A. Early landraces were assigned in general to genetic groups K7 and K9, with K7 landraces being shorter. As for late flowering landraces, they were assigned mainly to K1, K2, and K4, landraces from K4 being taller than the others. Admixed landraces were also found among the extreme phenotypes for flowering or plant height. In general, all genetic groups showed large variability in both flowering time and plant height. Interestingly, landraces assigned to K3 (South–Central Italian) were shorter and earlier than landraces from K5 (Northern Italian), which could be associated with specific uses or management practices. In addition, ASI was similar for all the genetic groups except for K8 (Northern Flint), which showed the highest ASI.

### 3.4. Principal Component Analysis 

The highest correlations among the traits studied were between flowering and plant (0.97) and ear height (0.89) while the lowest correlations were observed between ASI and the other traits, ranging from 0.17 for EPHR to 0.47 for DS.

Principal component analysis was carried out to summarize the information relating to the six traits evaluated. Only the first principal component (PRIN1) had an eigenvalue greater than 1, and this explained 72% of the variability. The second principal component (PRIN2), with an eigenvalue close to 1, explained 16% of the total variance. In total, 95% of the total variance was explained through the first three principal components (Table 6).

Days to flowering and plant architecture were the characteristics with the highest (positive) influence on PRIN1, while ASI presented the highest (positive) coefficient for the second principal component. This means that tall and late varieties showed high PRIN1 values while high PRIN2 values corresponded to high ASI (Table 6).

In the graphic representation of the PCA, landraces were positioned in all four quadrants, corresponding to substantial phenotypic variation in all genetic groups (Figure 5). The hybrid checks were mainly located in quadrant IV (positive PRIN1 and negative PRIN2 values) as they were tall and late flowering and presented small anthesis–silking intervals. For Axis 1, landraces assigned to K3 and K7 were placed mainly in the quadrants II and III, while landraces assigned to K1, K2, K4, and K5 were positioned in the quadrants I and IV. In reference to Axis 2, most varieties from Switzerland assigned to K1 were positioned in the two upper parts of the PCA chart (quadrants I and II).

### 3.5. Cluster Analysis 

Cluster analysis based on the phenotypic data of 588 landraces and five hybrid checks grouped them into 15 subclusters, which, in turn, were grouped into five main clusters A–E, with clusters B and E containing most landraces (Figure 6, Table 7 and Appendix A).

**Cluster A** was the smallest cluster, formed of only subcluster 1, containing two landraces characterized as very late and tall. These Spanish popcorn populations (ESP0200, Sangonera La Seca and ESP0115, Alcantarilla) were embedded in the K2 genetic group, but were considered admixed, as they shared ancestry with other genetic groups, including K1 (popcorn).

**Cluster B** was formed of 308 landraces with trait values slightly higher than the average landrace. Cluster B was formed of six subclusters (2, 3, 5, 7, 11, and 14). The closest subclusters 3 and 11 grouped 86 and 66 landraces, respectively, and contained the earliest flowering landraces of cluster B. In subcluster 3, more than half of the landraces were admixed (<0.6 probability of assignment to a particular genetic group). More than half of these admixed landraces had K4 as their nearest group (Corn Belt Dent germplasm), and ten landraces from Eastern Europe were also assigned to K4. Likewise, almost two thirds of the landraces in subcluster 11 were admixed, although many of them were related to K4. The remaining landraces in this subcluster were distributed across genetic groups. None of the landraces belonged to the ‘Caribbean’ group [33], while few of them were included in the Northern Flint group described in the same study. Subclusters 7 and 5, with 45 and 48 landraces, respectively, grouped the latest flowering landraces of cluster B. In both subclusters, more than half of the landraces belonged or were related to the K4 (Corn Belt Dent germplasm) genetic group. Subcluster 14 consisted of 19 intermediate flowering landraces. Genetically, most of the landraces were included in or related to groups K1 and K2 and came from Southeastern Spain, including popcorn landraces. Subcluster 2 consisted of 44 landraces, most of them from Switzerland, characterized by large ASI and ascribed, in a high proportion, to the K8 genetic group (Northern Flint germplasm). In summary, more than 60% of the populations belonging to groups K1, K2, K4 and K5 were included in phenotypic cluster B, while less than 15% of the populations belonging to group K3 were in this cluster. So, this phenotypic cluster encompassed Caribbean, Northern Italian, and Corn Belt Dent germplasms.

**Cluster C** consisted of 44 landraces; 29 belonging to subcluster 8 and 15 to subcluster 4. Cluster C was a group of late landraces with high insertion of the main ear. In subgroup 4, 14 out of the 15 landraces originated from Eastern or Southern Spain or the Canary Islands. Subcluster 4 included some popcorn landraces and was highly related to genetic groups K1 and K2. Subcluster 8 was formed of landraces from different origins that were highly related to the genetic group K4 (Corn Belt Dent).

**Cluster D**, representing only subcluster 6, was made up of eight early landraces, most of them being part of the K9 (Pyrenean) or K7 (Galician) genetic groups. They stood out, on average, for shedding silks before anthers and, consequently, presenting negative ASI values. 

**Cluster E** was made up of 227 early and short varieties (226 landraces and the earliest hybrid check F210) grouped into five subclusters: 13 (32 landraces), 15 (25 landraces), 12 (46 landraces), 10 (76 landraces), and 9 (47 landraces). Out of these 226 landraces, 114 were admixed, 15 belonged to K3 and 5 to K5 genetic groups (Italian landraces), 29 to K7 (Galician genetic group), 19 to K9 (Pyrenean genetic group), and 38 (mainly in sub-cluster 13) to K8 (Northern Flint group). Sub-clusters 12 and 10 were joined first, followed by sub-clusters 9, 15, and 13.

Clusters B and C joined to form a new group of late and tall landraces with large anthesis–silking intervals. Genetically, they belonged or were mostly related to the K4 genetic group, but also to the K1, K2, and K5 groups. They represented the European landraces with the greatest contribution from the Corn Belt Dent and Caribbean germplasms. Clusters D and E united to form a new group of early and short landraces with a high presence of landraces from the Northern Flint (K8), Galician (K7), and Pyrenean (K9) genetic groups. Clusters B–C and D–E finally joined cluster A, which was a small cluster formed by two popcorn landraces that stood out for being late and presenting remarkably high values for ear height and relative ear height.

## 4. Discussion

The EVA Maize Network enabled for the first time the systematic evaluation of a large collection of European maize germplasm in multilocation trials, which, together with the SNP genotyping of all accessions, provided an overview of the diversity of maize available in European genebanks. Previous studies on the European maize core collection (EUMLCC) only included accessions from six countries and thus captured only a limited range of European germplasm. In addition, characterization of the EUMLCC was carried out for specific traits in specific environments and was therefore not as extensive as that carried out in the EVA network.

The EVA maize collection presented in this study captured a large proportion of European maize diversity. Some landraces had been analyzed in previous studies using SNP [33], RFLP [6,45,47], SSR [48,49], or isozymes [1]. Seventy-five accessions in the EVA collection were also included in the EUMLCC and evaluated for various phenotypes in the RESGEN88 project. Despite not including German and Greek landraces from the original EUMLCC panel, the EVA collection extended the diversity to populations from Switzerland, Croatia, Romania, and Serbia, and included additional materials from France, Italy, Spain, and Portugal. This new study reveals genetic diversity that had not previously been represented in the EUMLCC and could be used to review the composition of the core collection. For example, we identified new, unique clusters in the genotypic grouping, which did not contain any accessions previously included in the EUMLCC. Genotyping data highlighted a clear clustering of accessions into nine genetic groups. The EVA collection contained 349 accessions that could be assigned to a prevalent group and 277 admixed accessions not assigned to any genetic group (membership <0.6 to any genetic group). The mixed genetic background of the latter is likely to have resulted from breeding activities since the introduction of maize to Europe [50]. The main lines of our results appear consistent with historical records and findings from previous molecular studies. Gauthier et al. [6] classified maize populations from Europe into three main clusters consistent with geographic origins: a Northeastern cluster, a Southeastern cluster with semi-dent populations from Southeastern Europe, and finally a Northwestern cluster with early flint populations from Northwestern Europe. They also detected a possible direction of gene flow indicating that Northern Flint and Caribbean populations were introduced to Northern and Southern Europe, respectively [51,52,53]. The large representation of diverse germplasm from several countries in the present study highlighted finer-scale patterns. 

The K1 molecular group, highlighted in the present research, is a special type of Mediterranean germplasm, composed mainly of popcorn accessions. These were probably genetically isolated from the rest of the Mediterranean populations due to the presence of the gametophytic factor in most popcorn populations [54]. Due to the long growth cycle of these materials, some of them were included in phenotypic cluster B because of their large plant size, while other accessions were in cluster C because of their late cycle and high ear insertion. Following this clustering, two populations from the FRA015 collection were re-investigated and validated as popcorn (M. Vincent, personal communication), showing how the present study can contribute to genetic resources characterization.

Group K2 could be explained through the first introductions of maize from the Caribbean area into Seville in 1494 [1] and subsequent introductions throughout the South of Spain. Though these first introductions were probably not very successful, their singularity is still present in that area. These varieties were included in phenotypic cluster A because they are late and tall, as would be expected from the photoperiod sensitivity related to their Caribbean origin [55]. This may also explain why some of these populations were included in phenotypic clusters B and C, with large, late flowering plants. Similarly, groups K3 (South-Central Italy) and K5 (Northern Italy) represent the diverse introductions of maize made throughout the Mediterranean area. Maize was introduced in Spain and Italy almost simultaneously from Central American sources, giving rise to singular types of germplasm that remained in both countries. Some of the ancient introductions also had large plants because of their tropical origins [56,57]. The proximity of populations from Italy to populations from Spain has been shown to represent one of five genetic groups in 67 European and American inbred lines [46]. Interestingly, populations from Southern Italy, which are genetically close to Spanish populations from Southeastern Spain (K2), were significantly earlier than those populations and also those of Northern Italy. This “switch” in phenological traits may be explained by the fact that farmers in Southern Italy selected for materials with shorter cycles to avoid drought and/or high summer temperatures. Additionally, short-cycle maize landraces provided optimal results in terms of crop rotation for second cropping after wheat cultivation typical in these areas [58]. Finally, a small group of landraces from the Adriatic coast of Croatia were strongly assigned to this genetic group. These Croatian landraces were also phenotypically distinct from the remaining landraces collected in the same region, matching the phenotypic characteristics of the K3 group. This may suggest direct introduction from Italy and isolation, possibly related to the same cultivation practices applied on the other side of the Adriatic Sea. 

Northern Flints (K8) were introduced during the early 16th century [2]. The genetic groups K6, K7, and K9 are likely to have derived from admixture between these Northern Flints and tropical introductions. This is consistent with the origin of populations in Southwest France [59]. Some of these accessions were included in the phenotypic cluster D because they were probably selected for adaptation to short growth cycles. However, a larger number of them were included in cluster E, representing the introduction of flint from North America along the Atlantic coast.

Finally, in the late 19th and 20th century with the release of improved varieties, maize was introduced continuously from America, mainly from the USA. Therefore, the presence of Corn Belt Dent (K4) was established later than other groups, mainly in Eastern European countries where late varieties with high yield were very successful. Accordingly, many populations from Eastern Europe belonged to the Corn Belt Dent genetic group (K4) and to phenotypic clusters B and C and showed larger plant development than accessions from other clusters.

The analysis of phenotypic data for flowering time and plant or ear height explained only partially the separation patterns of the genotypic clustering. This indicates that the genetic groups exhibit large phenotypic diversity for these traits. The genotypic study shows the phylogenetic relationships due to the origins of the populations, while the similarities between varieties in plant height or flowering dates reflect selection for adaptation to diverse European environments into which the genetic groups were introduced. Similar observations were also reported in other studies [60,61]. Other traits may be better correlated to the genetic groups, and analysis of other evaluations of the EVA collection will provide information on this hypothesis. Another possible explanation for the missing separation into groups is the use of BLUEs across multiple trial locations for the analysis of phenotypic diversity. Perhaps disaggregating data according to location would show clearer patterning in the phenotypic PCA correlating with genetic groups.

## 5. Conclusions

The EVA Maize collection described in this study highlights the large genotypic diversity available across large ecogeographic regions in Europe. This collection provides good material for breeding programs to tackle climate change, and some landraces have also been registered as conservation varieties, making them available for growers seeking unique characteristics for specialty food products [5]. Others are being used in participatory breeding programs, which adapt a holistic transdisciplinary approach aiming at valorization along the entire value chain and promoting cultural legacy and identity [62]. 

Although not all European regions are represented in this collection, the goal of the ECPGR EVA initiative is to further explore genebank holdings through extension of the EVA Maize Network to countries in Central and Eastern Europe. Additional evaluations for agronomic traits and stress tolerance will add further knowledge of the collection, enabling identification of new variations to be used in the development of pre-breeding materials [63,64,65,66,67]. Furthermore, connecting the results with other ongoing (Dromamed, MineLandDiv) and future projects will ensure effective exploitation of the genetic resources conserved in European genebanks.

The EVA Network connects genebank collections with research institutes and public and private sector breeders. With its unique approach and much of the joint work provided as in-kind contributions by partners, these networks have been set up as long-term initiatives to continue exploring genetic diversity conserved in genebanks or on farms and make material and information available to end users, breeders, and growers.

## Figures and Tables

**Figure 1 biology-13-00454-f001:**
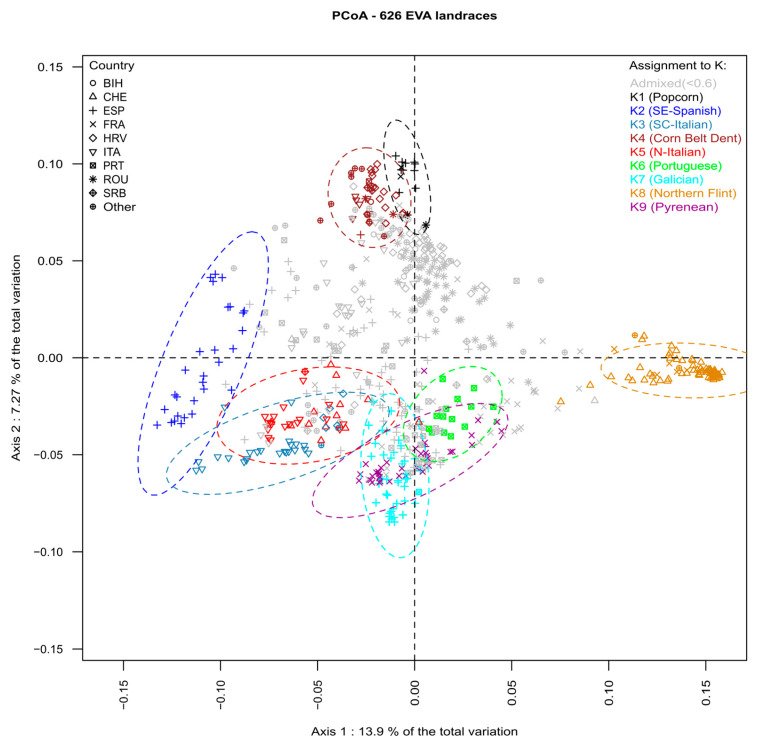
Principal coordinate analysis of 626 maize landraces based on modified Roger’s distance estimated with 23,412 SNP. Different colors represent landraces assigned (>0.6) to one of 9 genetic groups of ADMIXTURE. Symbols represent the country of origin of each landrace (BIH: Bosnia and Herzegovina, CHE: Switzerland, ESP: Spain, FRA: France, HRV: Croatia, ITA: Italy; PRT: Portugal, ROU: Romania, SRB: Serbia).

**Figure 2 biology-13-00454-f002:**
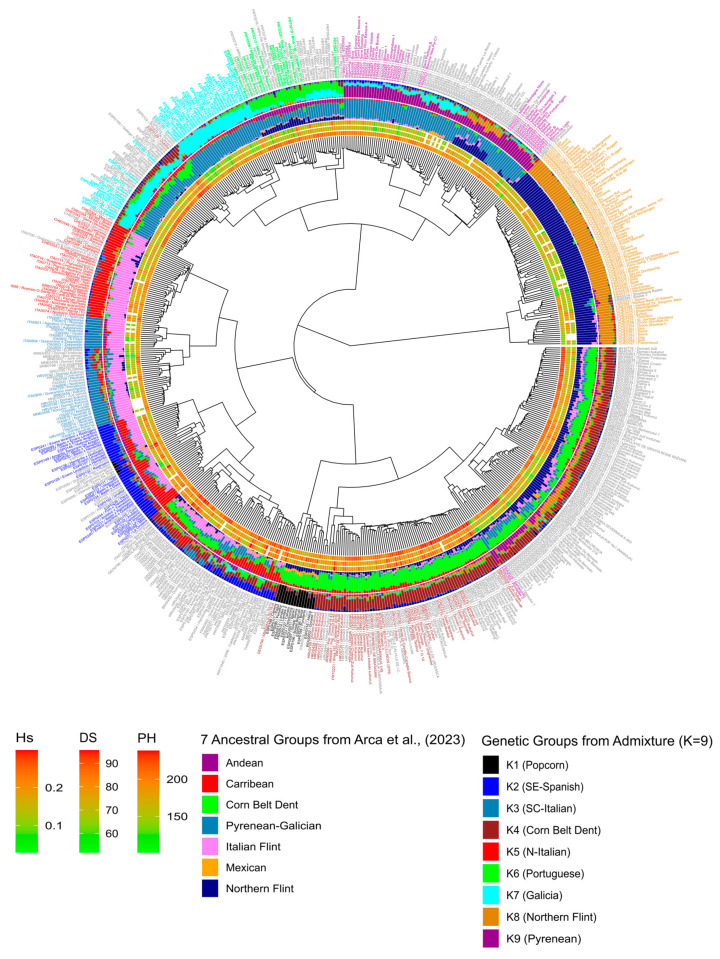
Dendrogram of genetic diversity based on modified Roger’s distance (MRD) of 626 landraces including additional data on (from internal to external layer): (i) expected heterozygosity (H_s_), (ii) female flowering time/days to silking (DS), and (iii) plant height (PH), with (iv) barplots from penalized regression to seven ancestral groups [33] and (v) barplots from ADMIXTURE analysis (K = 9). Individual accessions are labeled with the landrace code and common name, with label color corresponding to assignment to genetic groups (see also Appendix A).

**Figure 3 biology-13-00454-f003:**
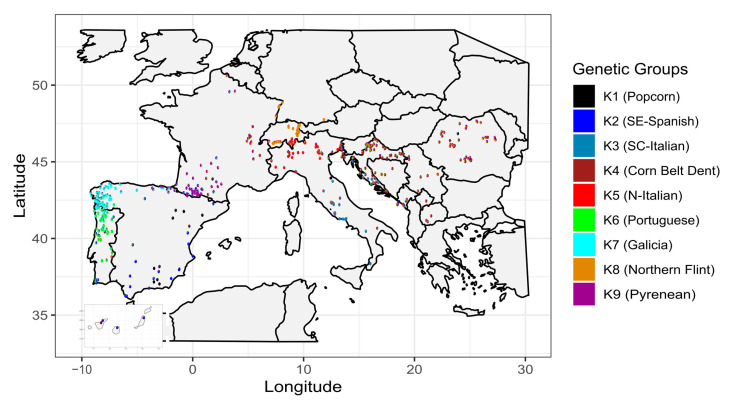
Map of collection sites of 483 EVA Maize accessions colored according to their membership of nine genetic groups. Each accession is represented by a pie chart where the color composition reflects the proportion of genome ancestry from nine genetic groups in the ADMIXTURE analysis (see legend).

**Figure 4 biology-13-00454-f004:**
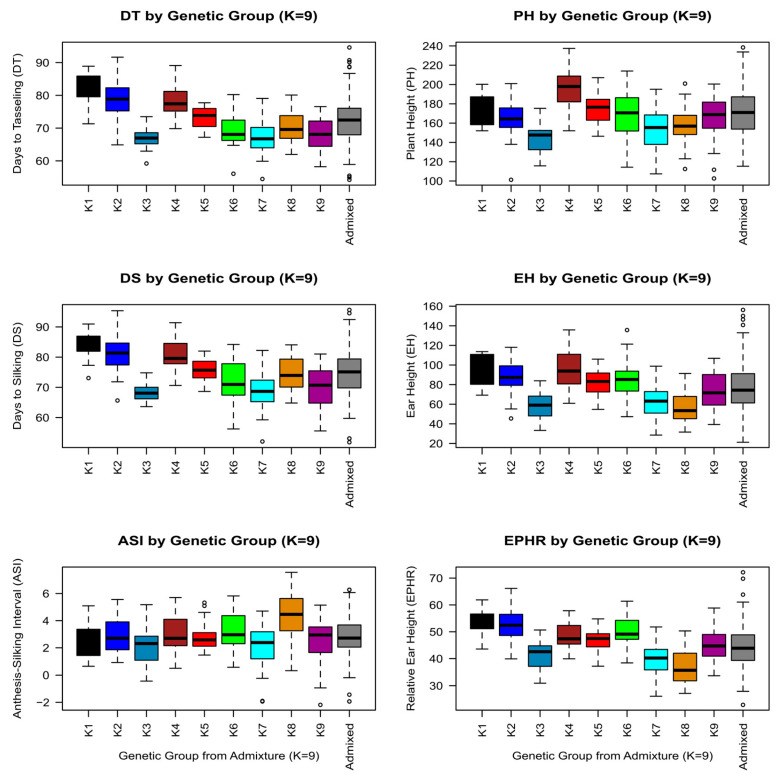
Boxplots of phenotypic BLUEs correlated to genetic grouping (K = 9), showing days to tasseling (DT), days to silking (DS), anthesis–silking interval (ASI), plant height in cm (PH), ear height in cm (EH), and relative ear height in % (EPHR) for landraces belonging to each genetic group K1 to K9 (>0.6) or admixed (<0.6), using the same color coding as other figures. Whiskers extend to minimum and maximum values (Q_0_, Q_4_), if not considered outliers.

**Figure 5 biology-13-00454-f005:**
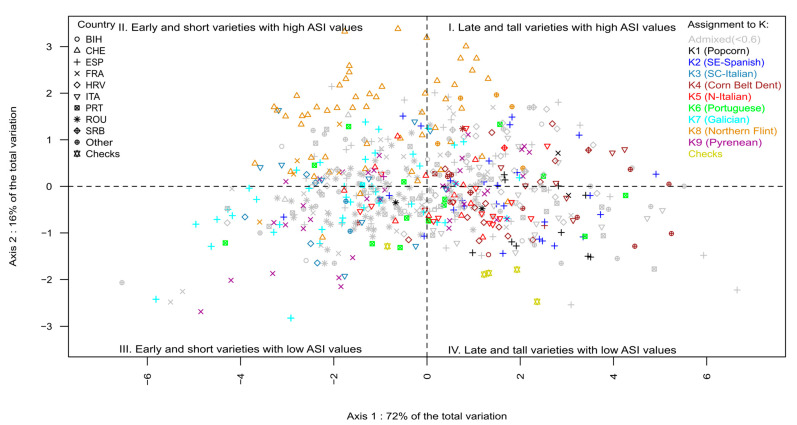
Representation of the first two components of the principal component analysis for phenotypic data collected from 588 maize landraces of the EVA collection. Different symbols represent country of origin ISO codes (BIH: Bosnia and Herzegovina, CHE: Switzerland, ESP: Spain, FRA: France, HRV: Croatia, ITA: Italy; PRT: Portugal, ROU: Romania, SRB: Serbia), while color represents the genetic structure of landraces (see legend). Hybrid checks are represented by both different symbols and colors. The first principal component (PRIN1) was positively related to plant architecture and days to flowering and the second principal component (PRIN2) was positively related to the anthesis–silking interval (ASI). The proportion of variance explained through each component is also shown.

**Figure 6 biology-13-00454-f006:**
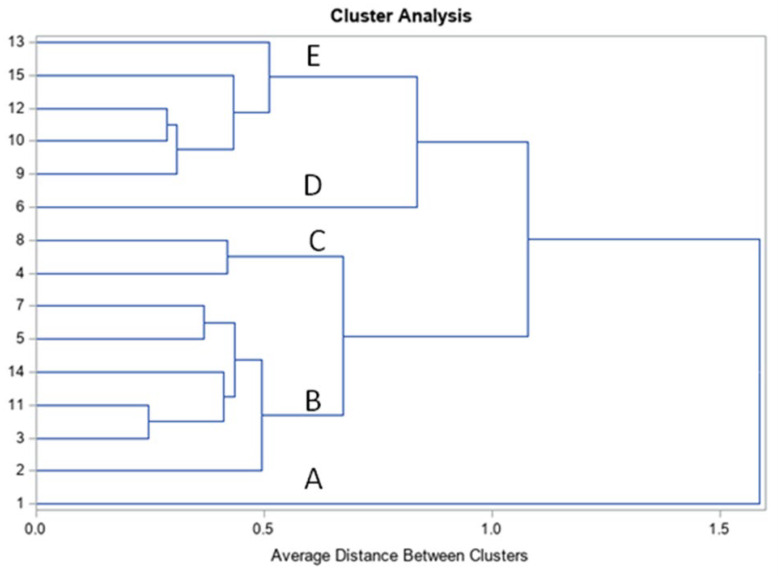
Dendrogram of pre-cluster and cluster analysis of 588 landraces and 5 hybrid checks based on phenotypic data. Euclidean distance and average linkage were used as a clustering method. The numbers on the left are the sub-clusters (see also Table 7).

**Table 1 biology-13-00454-t001:** Overview of the EVA Maize collection, including criteria for selection. Detailed passport data for all accessions are available in Appendix A.

Country	EVA Maize Genebank	Holding Institute Code (* WIEWS)	Number of Accessions	Part of ** EUMLCC	Criteria for Selection of Materials	References
Croatia	University of Zagreb, Faculty of Agriculture	HRV041	50	na	Representative of national collection based on a wide geographical distribution; phenological and morphological data.	[11]
France	INRAE—Montpellier	FRA015	80	16	Morphological traits to maximize number of classes; representative of national collection.	[7,12]
Italy	CREA-Cereal and Industrial Crops Bergamo	ITA386	65	19	Representative of different geographical areas in Italy; promising sources of stress-tolerance traits.	[8,13]
Portugal	INIAV, Braga	PRT001	42	17	Morphological, agronomical, and molecular characterization; previous knowledge from national breeding programmes.	[14,15]
	ESAC-IPC, Coimbra	PRT053	6	na	Drought tolerance; yield evaluation at different altitudes.	[16]
Romania	Suceava Genebank	ROM007	51	na	Early and semi-early (FAO groups 200–300); collected between 1973 and 2000; cold tolerance.	[17,18]
Serbia	Maize Research Institute Zemun Polje	SRB001	91	na	Drought tolerance; stability and high grain yield; good performance in breeding programmes.	[19,20]
Spain	Misión Biológica de Galicia, Pontevedra (MBG-CSIC)	ESP004	2	na	Representative of available diversity; corn borer resistance; good for breadmaking; cold and drought tolerance.	[1,9,21]
		ESP007	8	na		
		ESP009	137	23		
		ESP016	4	na		
		ESP119	4	na		
Switzerland	Agroscope, Nyon	CHE001	86	na	Representative of the collection from different regions in Switzerland; phenological and morphological data.	[22,23,24,25]
		**Total**	**626**	**75**		

* WIEWS, World Information and Early Warning System on Plant Genetic Resources for Food and Agriculture; ** EUMLCC, European Maize Landraces Core Collection; na, not applicable.

**Table 2 biology-13-00454-t002:** Trial locations for phenotypic evaluations in the EVA Maize network 2021–2023.

Country	Location	EVA Partner	Range of FAO Maturity Rating	# Accessions Evaluated	Years of Evaluation
**Croatia**	Zagreb	University of Zagreb	100–600	187	2022–2023
**France**	Ploudaniel	INRAE	100–800	598	2021–2023
**France**	Alzonne	KWS	300–500	326	2021–2023
**Germany**	Bernburg (Saale)	KWS	100–300	161	2021–2023
**Italy**	Bergamo	CREA—Cereal and Industrial Crops	200–500	178	2021–2023
**Italy**	Monselice	KWS	500–800	218	2021–2023
**Portugal**	Coimbra	ESAC	400–600	53	2022–2023
**Romania**	Suceava	Suceava genebank	100–300	108	2021–2022
**Serbia**	Belgrade	MRIZP	300–800	299	2021–2023
**Spain**	Pontevedra	MBG-CSIC	100–800	216	2021
**Switzerland**	Delley	Delley Semences et Plantes	100–250	84	2021–2022

**Table 3 biology-13-00454-t003:** Description of traits evaluated in EVA Maize phenotypic trials, following published descriptors [44].

Trait Name	Trait Acronym	Trait Description	Unit	Crop Ontology
**Days to tasseling (anthesis, male flowering)**	DT	IPGRI descriptor 4.1.1: number of days from sowing to when 50% of the plants have shed pollen.	d	CO_322:0000030
**Days to silking (female flowering)**	DS	IPGRI descriptor 4.1.2: number of days from sowing to when silks have emerged on 50% of the plants.	d	CO_322:0000031
**Anthesis–silking interval**	ASI	Time difference [in days] between anthesis and silking, calculated as (ASI = DS − DT).	d	CO_322:0000001
**Plant height**	PH	IPGRI descriptor 4.1.4: from ground level to the base of the tassel after milk stage, measured in cm.; observed value recorded from an average of 10 plants per plot.	cm	CO_322:0000994
**Ear height**	EH	IPGRI descriptor 4.1.5: from ground level to the node bearing the uppermost ear after milk stage, measured in cm.; observed value recorded from an average of 10 plants per plot.	cm	CO_322:0000017
**Relative ear height**	EPHR	Ear to plant height ratio calculated as (EH/PH) × 100.	%	

**Table 4 biology-13-00454-t004:** Genetic diversity parameters within the EVA panel and each genetic group (K = 9) based on highly assigned landraces (>0.6) from admixture analysis.

	Whole Panel	K1	K2	K3	K4	K5	K6	K7	K8	K9
	Mean ± S.D.	Mean ± S.D.	Mean ± S.D.	Mean ± S.D.	Mean ± S.D.	Mean ± S.D.	Mean ± S.D.	Mean ± S.D.	Mean ± S.D.	Mean ± S.D.
Size of the panel/group	626	14	29	31	40	32	14	43	75	39
Number of monomorphic SNPs	4	956	73	192	81	161	573	122	229	590
Allele number (A) at group level	2.00 ± 0.01	1.96 ± 0.2	2.00 ± 0.06	1.99 ± 0.09	2.00 ± 0.06	1.99 ± 0.08	1.98 ± 0.15	1.99 ± 0.07	1.99 ± 0.1	1.97 ± 0.16
Allele number (A) average within landraces	1.71 ± 0.13	1.57 ± 0.13	1.67 ± 0.09	1.69 ± 0.06	1.79 ± 0.08	1.65 ± 0.11	1.75 ± 0.05	1.72 ± 0.07	1.52 ± 0.11	1.67 ± 0.09
Minor allele frequency (MAF) at group level	0.25 ± 0.14	0.19 ± 0.15	0.21 ± 0.15	0.19 ± 0.15	0.25 ± 0.14	0.2 ± 0.15	0.21 ± 0.15	0.19 ± 0.15	0.16 ± 0.15	0.20 ± 0.16
Minor allele frequency (MAF) average within landraces	0.15 ± 0.03	0.13 ± 0.03	0.13 ± 0.02	0.14 ± 0.02	0.18 ± 0.02	0.14 ± 0.03	0.17 ± 0.01	0.16 ± 0.02	0.10 ± 0.02	0.15 ± 0.03
Total expected heterozygosity (H_t_) at group level	0.33 ± 0.15	0.27 ± 0.17	0.29 ± 0.16	0.26 ± 0.17	0.33 ± 0.15	0.27 ± 0.17	0.28 ± 0.17	0.27 ± 0.17	0.22 ± 0.17	0.27 ± 0.18
Expected heterozygosity (H_s_) average within landraces	0.21 ± 0.05	0.17 ± 0.04	0.19 ± 0.03	0.19 ± 0.02	0.24 ± 0.03	0.19 ± 0.04	0.23 ± 0.02	0.22 ± 0.03	0.14 ± 0.03	0.20 ± 0.03
Differentiation (G_st_) between landraces (within group H_t_)	0.35 ± 0.14	0.35 ± 0.14	0.36 ± 0.10	0.25 ± 0.09	0.26 ± 0.10	0.30 ± 0.13	0.17 ± 0.07	0.18 ± 0.10	0.36 ± 0.15	0.25 ± 0.13
Modified Roger’s distance (MRD) between landraces	0.24 ± 0.04	0.22 ± 0.04	0.23 ± 0.03	0.18 ± 0.03	0.21 ± 0.03	0.20 ± 0.03	0.16 ± 0.02	0.16 ± 0.02	0.20 ± 0.04	0.18 ± 0.04

**Table 5 biology-13-00454-t005:** Mean, range, and quartiles of 588 landraces and five check hybrids for field evaluation parameters.

	* DT	* DS	* ASI	* PH	* EH	* EPHR
**Landraces**						
Mean	72.3	74.8	2.9	169	76	44.3
Minimum	54.2	51.8	−2.2	101	21	22.9
Q25	67.5	69.6	2.0	152	59	39.2
Median (Q50)	72.2	75.1	2.8	169	74	44.2
Q75	76.2	79.5	3.9	185	90	48.9
Maximum	94.6	95.6	7.6	238	158	72.1
**Check hybrids**						
Mean	78.0	77.8	0.3	207	91	44.0
Check1 FAO210	68.2	68.6	0.9	199	74	37.4
Check2 FAO300	75.4	75.5	0.5	212	98	45.9
Check3 FAO400	78.6	78.4	0.3	205	92	45.0
Check4 FAO500	82.4	82.3	0.3	209	94	44.8
Check5 FAO600	85.4	84.3	−0.7	209	98	46.8

* Days to tasseling/anthesis (DT), days to silking (DS), anthesis–silking interval (ASI), plant height in cm (PH), ear height in cm (EH), and relative ear height in % (EPHR).

**Table 6 biology-13-00454-t006:** Coefficients of the different traits, Eigenvalues and explained variability for the first (PRIN1), second (PRIN2), and third (PRIN3) principal components.

* Traits	PRIN1	PRIN2	PRIN3
DT	0.450	−0.017	−0.445
DS	0.452	0.187	−0.370
ASI	0.180	0.924	0.176
PH	0.412	−0.124	0.730
EH	0.458	−0.211	0.254
EPHR	0.426	−0.224	−0.192
Eigenvalue	4.331	0.9723	0.3976
% variability explained	72	16	7

* Days to tasseling/anthesis (DT), days to silking (DS), anthesis–silking interval (ASI), plant height (PH), ear height (EH), and relative ear height (EPHR).

**Table 7 biology-13-00454-t007:** Means of clusters and sub-clusters of 588 landraces and five hybrid checks for field evaluation parameters.

Cluster	Sub- Cluster	N	* DT	* DS	* ASI	* PH	* EH	* EPHR
**A**		**2**	**92.4**	**93.9**	**1.95**	**210**	**153**	**71.08**
	1	2	92.4	93.9	1.95	210	153	71.08
**B**		**312**	**74.9**	**77.8**	**3.34**	**179**	**85**	**47.3**
	2	44	74.1	78.9	5.21	163	70	43.2
	3	86	73.6	75.1	3.28	182	86	47.0
	11	66	72.1	73.8	2.11	169	76	44.8
	14	19	74.9	77.8	3.34	179	85	47.3
	5	48	77.2	81.5	4.77	192	97	50.1
	7	49	77.6	79.5	2.31	195	99	50.6
**C**		**44**	**84.0**	**86.9**	**3.28**	**204**	**117**	**56.9**
	4	15	85.4	87.7	2.65	177	105	58.4
	8	29	86.9	90.9	4.44	227	136	57.8
**D**		**8**	**57.1**	**55.4**	**−1.32**	**121**	**34**	**31.2**
	6	8	57.1	55.4	−1.32	121	34	31.2
**E**		**227**	**66.7**	**68.7**	**2.50**	**150**	**56**	**37.8**
	9	47	66.4	67.0	0.95	156	62	40.0
	10	76	68.2	70.8	2.95	161	65	40.5
	12	46	66.6	68.6	2.46	139	49	36.4
	15	26	62.3	63.0	1.07	130	43	34.1
	13	32	66.9	71.3	4.86	147	46	33.3
**Total**		**593**	**72.3**	**74.8**	**2.94**	**169**	**76**	**44.3**

* Days to tasseling/anthesis (DT), days to silking (DS), anthesis–silking interval (ASI), plant height in cm (PH), ear height in cm (EH), and relative ear height as % (EPHR). N, number of accessions in each cluster/subcluster.

## Data Availability

Data generated within the EVA project are subject to an embargo period under the EVA cooperation agreement and will thereafter be made publicly available according to FAIR principles. Reasonable requests for access to raw data can be addressed to the corresponding authors.

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
