# Peer review of "Genetic and Phenotypic Evaluation of European Maize Landraces as a Tool for Conservation and Valorization of Agrobiodiversity"

_biology, 2024, doi:10.3390/biology13060454_

Round 1

Reviewer 1 Report

Comments and Suggestions for Authors

The authors in the manuscript titled “Genetic and phenotypic evaluation of European maize landraces as a tool for conservation and valorization of agrobiodiversity” evaluated the diversity pattern in 626 local landraces and traditional varieties collected from nine genebanks distributed across Europe in an attempt to assess its value in local adaptation to cater the challenges of climate change. The combination of high throughput genotyping with the geographic history of the landraces and the phenotypic traits make this a very interesting study. The research design is appropriate and analysis is comprehensive.  There are some minor comments for the authors.

The starting of abstract and introduction strongly focuses to introduce the project EVA which is good for the readers to understand the material but it might be good to focus more on the importance of diversity of the studied material under climate shift scenario. So, slight restructuring of Abstract and introduction in recommended.

In material and methods, using of DNA pool was described. DNA pool can cause inclusion of out-layers within a population/landrace. Please explain how the effect of these out-layers were minimized using this strategy.

In results, the picture quality of the figures needs to be improved. In figure 3, add abbreviations of names of countries and improve the picture quality.

The landraces used in this study were conserved ex situ in the genebanks. Discussion related to the current cultivation state of these landraces in areas of Europe would enrich the comprehension about the importance of genetic resource used in this study. As continues cultivation of these landraces in their native regions could have added more adaptability and distinct genetic pattern. So, addressing these points as future prospects in the discussion part will further improve the discussion section. 

Author Response

Dear Reviewer,

we thank you for your careful reading and useful suggestions to improve our manuscript.

We have addressed all comments in the revised manuscript and responded in the attached file. 

Thank you very much for your attention, best regards

Reviewer 2 Report

Comments and Suggestions for Authors

Dear authors

Your manuscript is very comprehensive, covering many aspects and elements of the genetic and phenotypic analysis related to the evaluation of a large number of landrace maize populations. Congratulations!

In the attached document, you find some consideration.

Author Response

(The authors gave the same response as above.)

Reviewer 3 Report

Comments and Suggestions for Authors

22.2.2024

The manuscript entitled "
Genetic and phenotypic evaluation of European maize landraces 2 as a tool for conservation and valorization of agrobiodiversity” was reviewed.

The manuscript delivers holistic morpho-molecular analysis of European maize genetic resources, mainly landraces. However, there are few major comments listed below. Therefore, I do not recommend the publication of this manuscript in "Biology" unless these corrections are made.

- The language is good. Nonetheless, “the” is being deployed in an annoying fashion!!! Please check the entire manuscript.

- Almost 70% of sentences are long (more than two lines) or very very long (three or four lines). This is not a scientific writing but rather a novel approach. Please shorten your sentences along the entire manuscript.

- The “Simple Summary” is completely repeated in “Abstract” please rewrite.

- The “Abstract” is very weak; the first four lines do not deliver any novel info. And the last eight lines are more like a descriptive rather than informative. You have plenty of real data that SHOULD be presented in the “Abstract”. Please correct accordingly.

- In line 53, you mentioned Germany as a source of maize genetic resources! But this is not true! See table (1). But rather was a trail location (see table 2). That was misleading!

- Figure 1: is good but with low resolution, please improve!

- Figure 2: this is the most important piece of information in this work. However, the resolution is very bad! Please use vector drawing with improved resolution. Or spread the layers in a linear frame rather than circular one.

- Figure 4: you need to indicate what do whiskers represent …..

- Figure 5: this is worst PCA I ever reviewed in my entire life. There is NO grouping at all. The plot of landraces is extremely dispersed.  The presented PCA1 and PCA2 are correlated with morphological traits that are highly influential by the environment (both locations and years)!  

- It is recommended to incorporate more recent articles as only 16 out 75 (ca. 21%) of cited articles were published in the last five years.

Comments on the Quality of English Language

- The language is good. Nonetheless, “the” is being deployed in an annoying fashion!!! Please check the entire manuscript.

- Almost 70% of sentences are long (more than two lines) or very very long (three or four lines). This is not a scientific writing but rather a novel approach. Please shorten your sentences along the entire manuscript.

Author Response

Dear Reviewer, 

we thank you for your careful reading and useful suggestions to improve our manuscript. We have addressed all comments in the revised manuscript and responded in the attached file. Thank you very much for your attention, best regards.
